# Synthesis, Biological Evaluation and Computational Studies of New Hydrazide Derivatives Containing 1,3,4-Oxadiazole as Antitubercular Agents

**DOI:** 10.3390/ijms232315295

**Published:** 2022-12-04

**Authors:** Daniele Zampieri, Sara Fortuna, Maurizio Romano, Alessandro De Logu, Gianluigi Cabiddu, Adriana Sanna, Maria Grazia Mamolo

**Affiliations:** 1Department of Chemistry and Pharmaceutical Sciences, University of Trieste, Via Giorgieri 1, 34127 Trieste, Italy; 2Italian Institute of Technology (IIT), Via E. Melen 83, 16152 Genova, Italy; 3Department of Life Sciences, University of Trieste, Via Valerio 28/1, 34127 Trieste, Italy; 4Department of Life and Environmental Sciences, Section of Microbiology, University of Cagliari, 09042 Monserrato, Italy; 5Department of Public Health, Clinical and Molecular Medicine, University of Cagliari, 09042 Monserrato, Italy

**Keywords:** tuberculosis, antimycobacterial, oxadiazole, hydrazide, InhA, MIC

## Abstract

To extend our screening for novel antimycobacterial molecules, we have designed, synthesized, and biologically evaluated a library of 14 new hydrazide derivatives containing 1,3,4-oxadiazole core. A variety of mycobacterial strains, including some drug-resistant strains, were tested for antimycobacterial activity. Among the compounds tested, five showed high antimycobacterial activity (MIC values of 8 μg/mL) against *M. tuberculosis* H37Ra attenuated strain, and two derivatives were effective (MIC of 4 µg/mL) against pyrazinamide-resistant strains. Furthermore, the novel compounds were tested against the fungal *C. albicans* strain, showing no antimycotic activity, and thus demonstrating a good selectivity profile. Notably, they also exhibited low cytotoxicity against human SH-SY5Y cells. The molecular modeling carried out suggested a plausible mechanism of action towards the active site of the InhA enzyme, which confirmed our hypothesis. In conclusion, the active compounds were predicted in silico for ADME properties, and all proved to be potentially orally absorbed in humans.

## 1. Introduction

Tuberculosis (TB) continues to be the most lethal infectious disease in the world despite intensive efforts over the past 20 years to create novel medications, diagnostics, and vaccines with expanding pipelines.

According to the WHO report TB, since the UN high level meeting (2018), more individuals have been treated for TB, with over 14 million people receiving care in 2018 and 2019 [1]. The number of people provided with TB preventive treatment has quadrupled since 2015, from 1 million in 2015 to over 4 million in 2019. Despite the fact that the COVID-19 pandemic threatened to reverse the progress made in previous years, a dramatic drop in TB case notifications brought on by the pandemic has been documented, with a worldwide decline of 18% between 2018 and 2020 [1,2].

The disease’s primary cause, *Mycobacterium tuberculosis*, is responsible for more deaths than any other single infectious pathogen on a worldwide scale. The present treatment strategy involves a long-term polypharmacology approach, which frequently leads to an increase in *M. tuberculosis* multidrug-resistant strains (MDR). Despite the recently FDA-approved drugs Bedaquiline and Delamanid, there is yet a pressing need for novel antimycobacterial treatments with a clear mechanism of action and low toxicity, particularly against emerging MDR strains.

Among several mycobacterial enzymes (i.e., alanine-racemase [3], metioninaminopeptidase (MetAP) [4], isocitrate lyase (ICL) [5], menaquinone-B (MenB) [6], decaprenyl-phosphoryl-β-D-ribose oxidase (DprE1) [7] and others), 2-*trans*-enoyl-acyl carrier reductase (InhA) is still a crucial target for the development of novel drugs. InhA encodes for a protein with NADH-specific enoyl-acyl carrier protein (ACP) reductase activity which converts 2-unsaturated to saturated fatty acids and is involved in the elongation of long-chain fatty acids to mycolic acids [8]. Isoniazid (INH), a first line (pro-)drug used for many years to treat TB, has as its main molecular target this protein. One of the really potent tools used in the development of novel drugs is the hybridization strategy, which is based on the generation of a new chemical entity from the combination of two or more bioactive pharmacophore scaffolds [9,10].

The oxadiazole ring is present in numerous compounds gifted with a wide range of pharmacological activity, including anticancer, antialgic, antiviral, antihypertensive, anticonvulsant, antifungal, and antibacterial properties (including antimycobacterial) [11]. On the other hand, several antimycobacterial compounds contain a hydrazone, hydrazide or hydrazine function [12]. The 1,3,4-oxadiazole ring can represent a cyclic form of the INH hydrazide motif, as well as the effective biologically active ring. To enhance the lipophilicity of the synthesized compounds, the oxadiazole nucleus has been linked, through a hydrazone moiety, to several aromatic rings, including heterocycles. Herein, we designed and synthesized a series of new hydrazide derivatives **1a–n**, containing 1,3,4-oxadiazole unit (Figure 1) and tested them for antimycobacterial activity towards several mycobacterial strains. 

## 2. Results and Discussion

### 2.1. Chemistry

The synthesis of the final 1,3,4-oxadiazole-hydrazone derivatives is depicted in Figure 1. The synthesis of the title compounds **1a–n** starts from isonicotinoyl hydrazide or benzoyl hydrazide, which were reacted with ethyl chloroglyoxilate and *p*-TosCl to give the ethyl 5-aryl-1,3,4-oxadiazole-2-carboxylate intermediates **2a,b**. The latter was treated with an excess of hydrazine hydrate in ethanol, at reflux temperature, to afford the corresponding carbohydrazides **3a,b**. The last step includes several aromatic and heterocyclic aldehydes, through a conventional Schiff reaction, to produce the final 2,5-disubstituted 1,3,4-oxadiazole compounds **1a–n** (Figure 1). 

All the final derivatives were obtained as solids and were characterized by melting points, spectroscopic techniques such as IR, ^1^H-NMR, ^13^C-NMR, LRMS spectrometry and elemental analysis and were found to match the predicted structures. 

The ^1^H-NMR spectra of all the final compounds were consistent with the presence of pure Z geometric isomers. Only one derivative (**1j**) showed an equal quantity of both Z and E isomers, and from literature data, the signals of the hydrazide and azomethine protons of the E isomers of hydrazido-hydrazone derivatives and hydrazone derivatives, appear at lower fields with respect to the corresponding signals of the Z isomers [13]. These results led us to assign the Z configuration to the isomers that made up a higher percentage of the mixtures.

### 2.2. Antimycobacterial Activity 

All the new 1,3,4-oxadizole-hydrazone hybrid compounds **1a–n** were evaluated for their in vitro antimycobacterial activity against *M. tuberculosis* H37Ra attenuated strain, H37Rv virulent strain and several resistant strains as reported in Table 1 and Table 2. All compounds were assessed for their minimal inhibitory concentrations (MICs), which are calculated as the lowest concentration of tested substance needed to totally block bacterial growth. The MIC values (μg/mL and μM for the active compounds) of all the newly synthesized compounds **1a–n** (and reference drug INH, as a standard) were determined in triplicate (Table 1 and Table 2).

The results showed that all the 5-(pyridine-4-yl)-1,3,4-oxadiazole subseries **1a–g** exhibited weak antimycobacterial activity (32 > MIC > 64 μg/mL), while the 5-phenyl substituted oxadiazole subseries **1h–n** produced interesting findings. In fact, when the compounds present an unsubstituted monocyclic aromatic ring (Ar), linked through the hydrazide group in position 2 of the oxadiazole unit, or a heterocycle moiety, the activity increased with MIC values of 8 μg/mL as for **1h**, **1k–n**. The *para*-substitution on the aromatic portion (Ar), with available CH_3_ and Cl substituents, leads to a decrease in activity, in particular for the subseries **1a–g** (i.e., **1b** MIC > 64 μg/mL). In general, geometric isomerism (Z/E) may have an impact on the activity as different isomers might occupy different sites in the receptor pocket. In this case, a computational analysis (data not reported for compound **1h**) showed that the predicted binding affinity of the compound towards the examined target (InhA) is very close to that of the two isomers and is significantly beyond the capability of a scoring function to discriminate between the two geometric isomers. Since only one derivative of the series showed the presence of an equal quantity of the two geometric isomers Z and E, and moderate antimycobacterial activity (**1j**, MIC = 16 μg/mL), we opted against further chemical investigation on this molecule.

All the compounds were also tested against some resistant strains of mycobacteria and the virulent *M. tuberculosis* H37Rv strain. Data are reported in Table 2.

The most effective compounds shared a similar profile to the H37Rv strain, with a MIC range of 8 μg/mL for compounds **1k** and **1l**. The same compounds are also the best candidates against pyrazinamide-resistant strain (MIC 4 μg/mL). On the contrary, the INH-resistant strain was virtually unaffected by any of the tested derivatives. Moreover, a strain of *C. albicans* (ATCC90028) was used to test the whole series, and none of the novel derivatives showed inhibition activity at 100 μg/mL concentration, demonstrating a good selectivity for the mycobacteria.

### 2.3. Cytotoxicity Study

We also evaluated the cytotoxicity (CC_50_) of the most active compounds toward SH-SY5Y cells. The results (Appendix A) showed that compounds **1h** and **1n** had no cytotoxicity at tested concentrations, while compound **1k** exhibited slight cytotoxicity of 50 μM. The results confirmed the hypothesis that the derivatives have a good safety profile.

### 2.4. Molecular Modelling

Several studies reported the interaction of oxadiazole nucleus with InhA mycobacterial protein and various oxadiazole-based derivatives have been synthesized for their antitubercular activity [14,15,16]. Thus, by using InhA mycobacterial enzyme as a substrate, we computationally examined the interaction of 2 out of 14 novel compounds to confirm the potential mechanism of action of our derivatives. We selected one derivative per subseries, which is **1a** and **1h**.

We docked the two compounds into the InhA pocket. The lowest energy conformations were found in the target pocket with overlapping conformations for both compounds (Figure 2a). A careful analysis of the poses (Figure 2b,c) already allowed us to rationalize the experimental observation as both **1a** and **1h** had a shape that nicely matched the available pocket, but **1a** lower affinity for the substrate could be evinced by analyzing the target pocket pharmacophore features in comparison with the substrate’s chemistry. Indeed, the introduction of a nitrogen atom into the benzene ring polarizes it, which is expected to be inconvenient from the pharmacophore model (Figure 2d–f). Indeed, Phe41 interacts with the ring by π-π stacking (Figure 2f). Other features were appropriately located, and we note that there is room for further functionalization (Figure 2d–f). Overall, molecules are kept in place thanks to 10 hydrophobic interactions (Figure 2g). All substrates formed a hydrogen bond with the backbone nitrogen atom of Gly96 through their carbonyl oxygen atom (Figure 2g). Subsequent molecular dynamics simulations further confirmed these observations.

The analysis of the 250 ns long molecular dynamics (MD) simulations of each complex (Figure 3) showed that while the InhA backbone did not undergo large rearrangements along the simulated time (its root mean squared deviation, RMSD, stayed below 0.5 nm in both systems), only compound **1a** maintained its position inside the pocket (Figure 3a,c). The erratic behavior of compounds **1h** and RMSD with respect to the receptor backbone was mirrored by its predicted affinity and is in contrast with that measured for compound **1a**. Indeed, along the MD trajectory, we re-scored the complex with the same scoring function used for generating the docked poses (Figure 3c,d). In particular, the comparison among the distributions of the scores confirmed derivative **1h** to be the lowest scoring ligand as (i) it reached scores as low as −12 kcal/mol (Figure 3d), (ii) the distribution of the scores along the trajectory was sharp, and (iii) its median at −9.5 kcal/mol was lower than that calculated for compound **1a**. Finally, end simulation snapshots (Figure 3e,f) clearly showed that **1a** underwent major rearrangements, while compound **1h** tended to rearrange its 1,3,4-oxadiazole ring to better fit the pharmacophore (compare Figure 2a with Figure 3f).

Hence, the identified compound’s ability to fit the protein pocket was computationally confirmed. The pharmacophore model (Figure 2b–c) revealed room for further functionalization. In particular, the introduction of the benzene ring with a hydrogen bond acceptor could lead to a ligand better suited for the identified pocket. Another option would be to cyclize the molecule by linking the hydrogen bond acceptor group to an aliphatic linker connecting to the other benzene ring, creating a macrocycle capable of blocking the large pocket entrance of InhA.

### 2.5. In Silico Pharmacokinetic Parameters

With the help of the SwissADME tool (www.swissadme.ch, accessed on 15 May 2021), we evaluated the five most promising compounds for a prediction of the drug-likeness properties in silico [17] with the most common pharmacokinetic parameters, on the basis of the extended version of Lipinski’s rule of five (RO5) [18]. The RO5 extended criterion means that an orally active drug should not violate more than one of the following requirements: MW ≤ 500; HBA and HBD (related to the membrane permeability) ≤10 and ≤5, respectively; logP and logS (related to the intestinal absorption) ≤ 5; PSA ≤ 140 Å. All the evaluated compounds **1h** and **1k–n**, in comparison with INH as references standard, exhibited good drug-likeness properties being all the values within the ranges of RO5, suggesting that these derivatives would be orally absorbed in humans (Table 3).

## 3. Materials and Methods

### 3.1. Chemistry

#### 3.1.1. Chemical Reagents and Instruments

Commercially available chemicals were of reagent grade and used as received with the exception of CH_2_Cl_2_ which was distilled over anhydrous CaCl_2_ under an argon atmosphere. Reaction courses and product mixtures were routinely monitored by thin-layer chromatography (TLC) on silica gel precoated F_254_ Merck plates. Celite 545 was used for the filtration of the Pd/C catalyst. Melting points were determined with a Stuart SMP300 apparatus and are uncorrected. Infrared spectra were recorded on a Jasco 4700 FTIR spectrophotometer equipped with an ATR device. Nuclear magnetic resonance spectra were determined on a Varian 400 MHz (400 MHz for ^1^H-NMR and 101 MHz for ^13^C-NMR). Chemical shifts are reported as δ (ppm) in CDCl_3_, DMSO-*d_6_*, DMF-*d_7_* or C_5_D_5_N related to tetramethylsilane employed as the internal standard; one drop of D_2_O was added to assign NH protons. Coupling constants (J) are reported in Hz and the splitting abbreviations used are s, singlet; d, doublet; dd, doublet of doublets; dd, doublet of doublet of doublets; t, triplet; dt, doublet of triplets; td, triplet of doublets; q, quartet; m, multiplet; br, broad. Microanalyses (C, H, N) were carried out with Elementar Vario ELIII apparatus and were in agreement with theoretical values ± 0.4%. LR-MS (ESI) spectra were obtained on a Bruker Daltonics Esquire 4000 spectrometer by infusion of a solution of the sample in MeOH (HPLC grade).

#### 3.1.2. General Procedure for the Synthesis of Intermediates **2a,b**


*Ethyl 5-(Pyridine-4-yl)-1,3,4-oxadiazole-2-carboxylate*
**2a**


Three grams (0.0218 mol) of isonicotinoyl hydrazide (INH) were dissolved in 150 mL of CH_2_Cl_2_ at 0 °C in an ice bath. Triethylamine (4.5 g, 0.0436 mol, 2 eq) and ethylchloroglyoxylate (3.3 g, 0.0239 mol, 1.1 eq) were then added and the reaction mixture was left at rt for 8 h, monitored by TLC (CH_2_Cl_2_-EtOH 9:1). At reaction completion, 2.25 g (0.0218 mol, 1 eq) of triethylamine and 4.156 g (0.0218 mol, 1 eq) of *p*-TosCl were added and the reaction was left to stir for additional 4 h. The organic phase was washed with distilled water (3 × 100 mL), then with NaHCO_3_ sat. solution (1 × 100 mL) and finally with Brine (1 × 100 mL). The collected organic phase was dried over anhydrous MgSO_4_, filtered and evaporated to dryness. The solid residue was crystallized from abs. EtOH.

Light yellow solid. MW: 219.20; Yield: 4.17 g, 87% Mp: 84–86 °C; IR (cm^−1^): 1735. ^1^H NMR (400 MHz, CDCl_3_) δ 8.90–8.84 (m, 2H, arom.), 8.04–7.98 (m, 2H, arom.), 4.57 (q, J = 7.1 Hz, 2H, CH_2_), 1.49 (t, J = 7.1 Hz, 3H, CH_3_). ^13^C NMR (101 MHz, CDCl_3_): δ 164.58, 157.08, 154.03, 151.11, 129.89, 120.76, 63.88, 14.07.

With the same procedure, but with the use of benzoyl hydrazide as starting reagent, compound **2b** was obtained.


*Ethyl 5-phenyl-1,3,4-oxadiazole-2-carboxylate*
**2b**


Light yellow solid. MW: 218.21; Yield: 83%; Mp: 61–62 °C (70–71 °C, ref. [19]); IR (cm^−1^): 1751. ^1^H NMR (400 MHz, CDCl_3_) δ 8.09–7.98 (m, 2H, arom.), 7.56–7.35 (m, 3H, arom.), 4.46 (qd, J = 7.1, 1.3 Hz, 2H, CH_2_), 1.39 (td, J = 7.1, 1.3 Hz, 3H, CH_3_). ^13^C NMR (101 MHz, CDCl_3_): δ 166.30, 156.42, 154.30, 132.72, 129.16, 127.47, 122.66, 63.42, 14.02.

#### 3.1.3. General Procedure for the Synthesis of Intermediates **3a,b**


*5-(pyridin-4-yl)-1,3,4-oxadiazole-2-carbohydrazide*
**3a**


An ethanolic solution (50 mL) of **2a** (0.51 g, 2.3 mmol) was treated with an excess of hydrazine hydrate (0.46 g, 93 mmol, 4 eq) at reflux temperature for 6h. Upon cooling the solid formed was collected by filtration resulting in a chromatographically pure solid.

Light yellow solid. MW: 205.18; Yield: 0.22 g, 48% Mp: 209–211 °C; IR (cm^−1^): 3308, 3247, 3177, 1679; ^1^H NMR (400 MHz, DMSO-*d_6_*) δ 10.73 (br, 1H, NH), 8.85 (d, J = 5.0 Hz, 2H, arom.), 8.10–7.87 (m, 2H, arom.), 4.85 (br, 2H, NH_2_). ^13^C NMR (101 MHz, DMSO-*d_6_*): δ 164.03, 163.60, 159.00, 151.48, 130.49, 121.00.

With the same procedure compound **3b** was obtained.


*5-phenyl-1,3,4-oxadiazole-2-carbohydrazide*
**3b**


Light yellow solid. MW: 204.19; Yield: 55% Mp: 175–177 °C (70–71 °C, ref. [19]); IR (cm^−1^): 3284, 1751; ^1^H NMR (400 MHz, DMSO-*d_6_*) δ 8.06 (d, J = 7.4 Hz, 2H, arom.), 7.62 (dq, J = 14.4, 7.2 Hz, 3H, arom.), 6.61 (br, 2H, NH_2_). ^13^C NMR (101 MHz, DMSO-*d_6_*): δ 165.08, 158.45, 152.57, 133.00, 129.92, 127.46, 123.22.

#### 3.1.4. General Procedure for the Synthesis of Final Compounds **1a–n**


*(Z)-N’-benzyliden-5-(pyridin-4-yl)-1,3,4-oxadiazol-2-carbohydrazide*
**1a**


An amount of 0.235 g (124 mmol) of **3a** was dissolved in 50 mL of abs. EtOH, then 0.131 g (124 mmol) was added with a few drops of AcOH and the reaction was stirred at reflux temperature for 6 h. Upon cooling the solid formed was collected by filtration resulting chromatographically pure.

Yellow solid; MW: 293.28; Yield: 0.33 g, 91%; Mp: 207–209; Rf (CH_2_Cl_2_-EtOH 9:1): 0.75; IR (cm^−1^, ATR): 3274 (NH) 1689 (C=O). ^1^H NMR (400 MHz, DMSO-*d_6_*): δ 12.79 (s, 1H, NH), 8.86 (d, J = 5.2 Hz, 2H, arom.), 8.60 (s, 1H, =CH), 8.05–7.95 (m, 2H, arom.), 7.78–7.69 (m, 2H, arom.), 7.47 (dp, J = 3.6, 1.8 Hz, 3H, arom.). ^13^C NMR (101 MHz, DMSO-*d_6_*): δ 164.19, 158.96, 151.52, 151.48, 149.96, 134.10, 131.23, 130.37, 129.38, 127.96, 121.07. MS-ESI: [M + H]^+^ = 294; elemental analysis calcd (%) for C_15_H_11_N_5_O_2_: C 61.43, H 3.78, N 23.88; found: C 61.20, H 3.62, N 24.10.


*(Z)-N’-(4-chlorobenzyliden)-5-(pyridin-4-yl)-1,3,4-oxadiazol-2-carbohydrazide*
**1b**


Light brownish solid; MW: 327.73 g/mol; Yield: 74%; Mp: 236–238 °C; Rf (CH_2_Cl_2_-EtOH 9:1): 0.73; IR (cm^−1^, ATR): 3239 (NH) 1684 (C=O). ^1^H NMR (400 MHz, DMSO-*d_6_*): δ 12.93 (s, 1H, NH), 8.90 (s, 2H, arom.), 8.59 (s, 1H, =CH), 8.04 (s, 2H, arom.), 7.81–7.73 (m, 2H, arom.), 7.58–7.44 (m, 2H, arom.). ^13^C NMR (101 MHz, C_5_D_5_N): δ 165.92, 160.80, 157.71, 152.65, 152.16, 137.84, 134.61, 131.73, 130.79, 130.67, 122.13. MS-ESI: [M + H]^+^ = 328, 330; elemental analysis calcd (%) for C_15_H_10_ClN_5_O_2_: C 54.97, H 3.08, N 21.37; found: C 55.20, H 3.35, N 21.05. 


*(Z)-N’-(4-methylbenzyliden)-5-(pyridin-4-yl)-1,3,4-oxadiazol-2-carbohydrazide*
**1c**


Light yellow solid; MW: 307.31 g/mol; Yield: 47%; Mp: 223–225 °C; Rf (CH_2_Cl_2_-EtOH 9:1): 0.71; IR (cm^−1^, ATR): 3223 (NH) 1680 (C=O). ^1^H NMR (400 MHz, CDCl_3_) δ 10.10 (s, 1H, NH), 8.89 (s, 2H, arom.), 8.29 (s, 1H, =CH), 8.07–8.00 (m, 2H, arom.), 7.75–7.68 (m, 2H, arom.), 7.25 (dd, J = 7.4, 1.7 Hz, 2H, arom.), 2.40 (s, 3H, CH_3_). ^13^C NMR (101 MHz, CDCl_3_) δ 164.79, 158.68, 151.81, 151.03, 149.22, 141.81, 140.43, 139.99, 130.22, 129.52, 128.16, 21.59. MS-ESI: [M + H]^+^ = 308; elemental analysis calcd (%) for C_16_H_13_N_5_O_2_: C 62.53, H 4.26, N 22.79; found: C 62.50, H 4.10, N 23.00.


*(Z)-N’-((1H-pyrrol-2-yl)methylen)-5-(pyridin-4-yl)-1,3,4-oxadiazol-2-carbohydrazide*
**1d**


Yellow solid; MW: 282.26 g/mol; Yield: 69%; Mp: 278–282 °C; Rf (CH_2_Cl_2_-EtOH 95:5): 0.36; IR (cm^−1^, ATR): 3308 (NH) 1689 (C=O). ^1^H NMR (400 MHz, DMSO-*d_6_*): δ 12.55 (s, 1H, NH), 11.66 (s, 1H, NH pyrr.), 8.89–8.81 (m, 2H, arom.), 8.40 (s, 1H, =CH), 8.05–7.94 (m, 2H, arom.), 6.96 (td, J = 2.7, 1.4 Hz, 1H, arom.), 6.55 (ddd, J = 3.8, 2.4, 1.5 Hz, 1H, arom.), 6.16 (dt, J = 3.5, 2.3 Hz, 1H, arom.). ^13^C NMR (101 MHz, DMSO-*d_6_*): δ 164.07, 159.15, 151.51, 149.30, 144.03, 130.44, 126.93, 124.01, 121.05, 115.38, 110.07. MS-ESI: [M + H]^+^ = 283, [M + Na]^+^ = 305; elemental analysis calcd (%) for C_13_H_10_N_6_O_2_: C 55.32, H 3.57, N 29.77; found: C 55.40, H 3.35, N 29.60.


*(Z)-N’-(pyridin-2-ylmethylen)-5-(pyridin-4-yl)-1,3,4-oxadiazol-2-carbohydrazide*
**1e**


White solid; MW: 294.28 g/mol; Yield: 69%; Mp: 255–259 °C; Rf (CH_2_Cl_2_-EtOH 95:5): 0.45; IR (cm^−1^, ATR): 3099 (NH) 1702 (C=O). ^1^H NMR (400 MHz, DMSO-*d_6_*): δ 13.07 (s, 1H, NH), 8.90–8.80 (m, 2H, arom.), 8.62 (s, 1H, =CH), 8.07–7.95 (m, 4H, arom.), 7.93–7.78 (m, 1H, arom.), 7.44 (ddd, J = 7.6, 4.8, 1.3 Hz, 1H, arom.). ^13^C NMR (101 MHz, DMSO *d_6_*) δ 164.27, 159.14, 158.83, 153.05, 151.52, 150.16, 150.10, 137.44, 130.32, 125.39, 121.07, 120.79. MS-ESI: [M + H]^+^ = 295; elemental analysis calcd (%) for C_14_H_10_N_6_O_2_: C 57.14, H 3.43, N 28.56; found: C 57.20, H 3.25, N 28.40.


*(Z)-5-(pyridin-4-yl)-N’-(pyridin-4-ylmethylen)-1,3,4-oxadiazol-2-carbohydrazide*
**1f**


Light brown solid; MW: 294.28 g/mol; Yield: 82%; Mp: 282–284 °C; Rf (CH_2_Cl_2_-EtOH 9:1): 0.42; IR (cm^−1^, ATR): 3142 (NH) 1706 (C=O).^1^H NMR (400 MHz, DMSO-*d_6_*): δ 13.11 (s, 1H, NH), 8.87 (d, J = 5.1 Hz, 2H, arom.), 8.67 (s, 2H, arom.), 8.59 (s, 1H, =CH), 8.05–7.99 (m, 2H, arom.), 7.68 (d, J = 4.9 Hz, 2H, arom.). ^13^C NMR (101 MHz, DMSO *d_6_*): δ 164.30, 158.79, 151.55, 150.85, 150.27, 149.10, 142.66, 141.21, 130.33, 121.10. MS-ESI: [M + H]^+^ = 295; elemental analysis calcd (%) for C_14_H_10_N_6_O_2_: C 57.14, H 3.43, N 28.56; found: C 56.80, H 3.32, N 28.83.


*(Z)-N’-(furan-2-ylmethylen)-5-(pyridin-4-yl)-1,3,4-oxadiazol-2-carbohydrazide*
**1g**


Brown solid; MW: 283.25 g/mol; Yield: 63%; Mp: 236–242 °C; Rf (CH_2_Cl_2_-EtOH 9:1): 0.5; IR (cm^−1^, ATR): 3226(NH) 1684 (C=O). ^1^H NMR (400 MHz, DMSO *d_6_*): δ 12.82 (s, 1H, NH), 8.89–8.83 (m, 2H, arom.), 8.46 (s, 1H, =CH), 8.04–7.95 (m, 2H, arom.), 7.89 (dd, J = 1.8, 0.7 Hz, 1H, arom.), 7.02 (dd, J = 3.5, 0.7 Hz, 1H, arom.), 6.65 (dd, J = 3.4, 1.8 Hz, 1H, arom.). ^13^C NMR (101 MHz, DMSO-*d_6_*): δ 164.20, 158.91, 151.51, 149.82, 149.31, 146.41, 140.71, 130.36, 121.06, 115.65, 112.89. MS-ESI: [M + H]^+^ = 284; elemental analysis calcd (%) for C_13_H_9_N_5_O_3_: C 55.13, H 3.20, N 24.73; found: C 55.40, H 3.45, N 24.45.


*(Z)-N’-benzyliden-5-phenyl-1,3,4-oxadiazol-2-carbohydrazide*
**1h**


White solid; MW: 292.29 g/mol; Yield: 52%; Mp: 194–196 °C; Rf (CH_2_Cl_2_-EtOH 95:5): 0.51; IR (cm^−1^, ATR): 3251 (NH) 1688 (C=O). ^1^H NMR (400 MHz, DMSO-*d_6_*): δ 12.77 (s, 1H, NH), 8.60 (s, 1H, =CH), 8.14–8.03 (m, 2H, arom.), 7.79–7.67 (m, 2H, arom.), 7.72–7.56 (m, 3H, arom.), 7.52–7.42 (m, 3H, arom.). ^13^C NMR (101 MHz, DMSO-*d_6_*): δ 165.69, 158.40, 151.24, 150.16, 134.16, 133.18, 131.17, 129.99, 129.37, 127.93, 127.59, 123.15. MS-ESI: [M + H] ^+^ = 293 [M + Na]^+^ = 315; elemental analysis calcd (%) for C_16_H_12_N_4_O_2_: C 65.75, H 4.14, N 19.17; found: C 65.55, H 4.30, N 19.34.


*(Z)-N’-(4-chlorobenzyliden)-5-phenyl-1,3,4-oxadiazol-2-carbohydrazide*
**1i**


White solid; MW: 326.74 g/mol; Yield: 65%; Mp: 243–245 °C; Rf (CH_2_Cl_2_-EtOH 9:1): 0.51; IR (cm^−1^, ATR): 3229 (NH) 1695 (C=O). ^1^H NMR (400 MHz, DMF-*d_7_*) δ 12.84 (s, 1H, NH), 8.74 (s, 1H, =CH), 8.19–8.11 (m, 2H, arom.), 7.92–7.83 (m, 2H, arom.), 7.78–7.64 (m, 3H, arom.), 7.62–7.47 (m, 2H, arom.). ^13^C NMR (101 MHz, DMF-*d_7_*) δ 165.69, 161,79, 158.44, 150.13, 149.54, 135.72, 133.27, 132.77, 129.61, 129.22, 129.17, 127.23, 123.24. MS-ESI: [M+H]^+^ = 327, 329; elemental analysis calcd (%) for C_16_H_11_ClN_4_O_2_: C 58.82, H 3.39, N 17.15; found: C 58.50, H 3.22, N 17.45.


*(E/Z)-N’-(4-methylbenzyliden)-5-phenyl-1,3,4-oxadiazol-2-carbohydrazide*
**1j**


White solid; MW: 306.32 g/mol; Yield: 47%; Mp: 200–202 °C; Rf (CH_2_Cl_2_-EtOH 9:1): 0.68; IR (cm^−1^, ATR): 3446 (NH) 1682 (C=O). ^1^H NMR (400 MHz, DMSO-*d_6_*): δ 12.71 and 12.24 (s, 1H, NH, 47/53 % E/Z), 8.55 and 8.56 (s, 1H, =CH, 47/53 % E/Z), 8.14–8.07 (m, 2H, arom.), 7.69–7.56 (m, 5H, arom.), 7.27 (t, J = 7.3 Hz, 2H, arom.), 2.33 e 2.34 (s, 3H, CH_3_, 47/53 % E/Z). ^13^C NMR (101 MHz, DMSO-*d_6_*): δ 158.43, 151.27, 150.07, 141.15, 140.99, 131.45, 131.11, 130.00, 129.96, 127.93, 127.86, 127.59, 123.16, 21.53. MS-ESI: [M + H]^+^ = 307; elemental analysis calcd (%) for C_17_H_14_N_4_O_2_: C 66.66, H 4.61, N 18.29; found: C 66.34, H 4.53, N 18.50.


*(Z)-N’-((1H-pyrrol-2-yl)methylen)-5-phenyl-1,3,4-oxadiazol-2-carbohydrazide*
**1k**


Yellow solid; MW: 281.27 g/mol; Yield: 45%; Mp: 185–187 °C; Rf (CH_2_Cl_2_-EtOH 9:1): 0.68; IR (cm^−1^, ATR): 3370 (NH) 1679 (C=O). ^1^H NMR (400 MHz, DMSO-*d_6_*): δ 12.48 (s, 1H, NH), 11.65 (s, 1H, NH pyrr.), 8.40 (s, 1H, =CH), 8.14–8.06 (m, 2H, arom.), 7.72–7.58 (m, 3H, arom.), 6.95 (td, J = 2.6, 1.4 Hz, 1H, arom.), 6.54 (ddd, J = 3.8, 2.4, 1.5 Hz, 1H, arom.), 6.15 (dt, J = 3.5, 2.3 Hz, 1H, arom.). ^13^C NMR (101 MHz, DMSO-*d_6_*): δ 165.57, 158.59, 149.54, 143.83, 133.11, 129.98, 127.54, 126.98, 123.91, 123.21, 115.26, 110.02. MS-ESI: [M+H]^+^ = 282; elemental analysis calcd (%) for C_14_H_11_N_5_O_2_: C 59.78, H 3.94, N 24.90; found: C 59.99, H 3.82, N 24.66.


*(Z)-5-phenyl-N’-(pyridin-2-ylmethylen)-1,3,4-oxadiazol-2-carbohydrazide*
**1l**


White solid; MW: 293.28 g/mol; Yield: 53%; Mp: 242–243 °C; Rf (CH_2_Cl_2_-EtOH 9:1): 0.75; IR (cm^−1^, ATR): 3197 (NH) 1707 (C=O). ^1^H NMR (400 MHz, DMSO-*d_6_*): δ 13.01–12.96 (m, 1H, NH), 8.62 (m, J = 3.5 Hz, 2H, =CH and arom.), 8.16–8.04 (m, 2H, arom.), 7.99 (d, J = 8.0 Hz, 1H, arom.), 7.88 (td, J = 7.7, 1.7 Hz, 1H, arom.), 7.68–7.58 (m, 3H, arom.), 7.43 (ddd, J = 7.5, 4.9, 1.2 Hz, 1H, arom.). ^13^C NMR (101 MHz, DMSO-*d_6_*): δ 165.77, 158.30, 153.13, 151.31, 150.40, 150.08, 137.43, 133.21, 129.99, 127.60, 125.34, 123.11, 120.76. MS-ESI: [M+H]^+^ = 294; elemental analysis calcd (%) for C_15_H_11_N_5_O_2_: C 61.43, H 3.78, N 23.88; found: C 61.67, H 3.58, N 23.56.


*(Z)-5-phenyl-N’-(pyridin-4-ylmethylen)-1,3,4-oxadiazol-2-carbohydrazide*
**1m**


White solid; MW: 293.29 g/mol; Yield: 49%; Mp: 243–245 °C; Rf (CH_2_Cl_2_-EtOH 9:1): 0.56; IR (cm^−1^, ATR): 3436 (NH) 1683 (C=O). ^1^H NMR (400 MHz, DMSO-*d_6_*): δ 13.03 (s, 1H, NH), 8.66 (d, J = 4.8 Hz, 2H, arom.), 8.57 (s, 1H, =CH), 8.09 (d, J = 7.4 Hz, 2H, arom.), 7.76–7.51 (m, 5H, arom.). ^13^C NMR (101 MHz, DMSO-*d_6_*): δ 165.79, 158.23, 150.82, 150.46, 148.81, 141.26, 133.23, 130.00, 127.60, 123.09, 121.70. MS-ESI: [M+H]^+^ = 294, [M+Na]^+^ = 316; elemental analysis calcd (%) for C_15_H_11_N_5_O_2_: C 61.43, H 3.78, N 23.88; found: C 61.40, H 3.66, N 23.95.


*(Z)-N’-(furan-2-ylmethylen)-5-phenyl-1,3,4-oxadiazol-2-carbohydrazide*
**1n**


Light brown solid; MW: 282.25 g/mol; Yield: 48%; Mp: 221–223 °C; Rf (CH_2_Cl_2_-EtOH 9:1): 0.49; IR (cm^−1^, ATR): 3243 (NH) 1687 (C=O).^1^H NMR (400 MHz, DMSO-*d_6_*): δ 12.75 (s, 1H, NH), 8.46 (s, 1H, =CH), 8.17–8.01 (m, 2H, arom.), 7.88 (dd, J = 1.8, 0.7 Hz, 1H, arom.), 7.74–7.53 (m, 3H, arom.), 7.01 (dd, J = 3.5, 0.7 Hz, 1H, arom.), 6.65 (dd, J = 3.5, 1.8 Hz, 1H, arom.).^13^C NMR (101 MHz, DMSO-*d_6_*): δ 165.70, 158.36, 150.04, 149.37, 146.34, 140.51, 133.17, 129.98, 127.57, 123.14, 115.50, 112.87. MS-ESI: [M+H]^+^ = 283; elemental analysis calcd (%) for C_14_H_10_N_4_O_3_: C 59.57, H 3.57, N 19.85; found: C 59.45, H 3.50, N 19.90.

### 3.2. Antimycobacterial, Antifungal and Cytotoxicity Studies

#### 3.2.1. Cultures Preparation

*Mycobacterium tuberculosis* H37Ra ATCC 25177, H37Rv ATCC 27294, INH-R ATCC35822, Pyr-R ATCC 35828, Rifa-R ATCC 35838 and Strepto-R ATCC 35820 were maintained on slants of Lowenstein–Jensen Medium (Difco, Becton Dickinson, Sparks, MD, USA), at 37 °C in an atmosphere of 5% CO_2_. An aliquot of these cultures transferred in Middlebrook 7H9 broth (Difco), supplemented with 10% ADC (albumin, dextrose, catalase, Difco) and 0.04% Tween 80 to avoid clump formation and incubated at 37 °C in 5% CO_2_. Cells were harvested by centrifugation, resuspended in saline until a 1 McFarland density (1 × 10^8^ cells/mL) was reached, diluted 1:20 in Middlebrook 7H9 broth with 10% ADC and 0.04% Tween 80, and sonicated in a bath-type sonicator to disrupt the clumps. The absence of clumps was also verified by a Zhiel–Neelsen staining. In order to confirm the title of inoculum employed, the bacterial suspensions were appropriately diluted and seeded on plates of Middlebrook 7H11 agar supplemented with 10% OADC (oleic acid, albumin, dextrose, catalase, Difco). The plates were incubated at 37 °C for 28 days and then determined by the number of colonies developed.

#### 3.2.2. Staging of the Solutions of the Test Compounds

The test compounds were dissolved in DMSO at a concentration of 1–5 mg/mL, depending on their solubility and stored at −20°C until use. Compounds were subsequently diluted in Middlebrook 7H9 broth with ADC 10% to obtain final concentrations of the testing compounds ranging between 64 and 0.03 μg/mL.

#### 3.2.3. Determination of Antimycobacterial Activity

MICs of the tested compounds were determined against *Mycobacterium tuberculosis* H37Ra ATCC 25177, H37Rv ATCC 27294, INH-R ATCC35822, Pyr-R ATCC 35828, Rifa-R ATCC 35838 and Strepto-R ATCC 35820 by the resazurin microtiter assay as described in ref. [20,21] with slight modifications; 100 μL of each solution of the tested compounds prepared as described above, were transferred into individual wells of 96-well plates and then 100 μL of the suspensions of mycobacteria previously arranged were added, in order to obtain a range of concentrations of the test compounds between 0.03 and 64 μg/mL. Each concentration was assayed in quadruplicate. The plates were incubated at 37 °C in 5% CO_2_ for 7 days; 30 μL of resazurin (Sigma-Aldrich Co., St. Louis, MO, USA) solution prepared at 0.01% (wt/vol) in distilled water, filter sterilized and stored at 4 °C, was added to each well, incubated 24–48 h at 37 °C, and assessed for color development. The change in color from blue to pink indicates bacterial growth. Therefore, the minimum inhibitory concentration (MIC) was attributed to the lower concentration of the test compound which inhibited the color change of resazurin. Isoniazid (INH, Sigma-Aldrich, St. Louis, MO, USA) was used as a reference compound. Each determination was repeated twice under the conditions described.

#### 3.2.4. Determination of Antifungal Activity

Minimal inhibitory concentration (MIC) was determined by the broth microdilution assay according to the Clinical and Laboratory Standard Institute reference document M27-A3. As test medium, RPMI 1640 medium (Sigma-Aldrich, St. Louis, MO, USA) without sodium bicarbonate, supplemented with L-glutamine (Gibco, Life Technologies, Grand Island, NY, USA) and buffered with 0.165 M MOPS (Sigma-Aldrich, St. Louis, MO, USA) at pH 7.0 was used. Two-fold dilutions of the testing compound at a concentration ranging between 200 and 0.39 μg/mL were obtained in the test medium and 100  µL of each dilution was dispensed into the wells of a 96-well plate; 100 μL inoculum of *C. albicans* ATCC 90028 were added to each well to obtain final concentrations of the testing compound ranging between 100 and 0.19 μg/mL and a final inoculum of 1 × 10^4^ cells/mL. Plates were incubated at 35 °C for 24–48 h and results were read visually. MIC was determined as the lowest concentration at which a 100% inhibition of growth compared with drug-free control was observed.

#### 3.2.5. Cytotoxicity

Cell viability was assayed by using Resazurin (SERVA Electrophoresis GmbH, Heidelberg, Germany). This cell-permeable dye is reduced by aerobic respiration of metabolically active cells to resorufin, whose fluorescence is proportional to the number of live cells [22,23]. The stock solution of resazurin sodium salt (440 μM, 10×) was resuspended in Phosphate Buffered Saline (PBS) and stored at −20 °C. Working solution (44 μM) was prepared on the same day of each assay by diluting resazurin stock solution in a standard culture medium. The human neuroblastoma cell line SH-SY5Y was used to test the cytotoxicity of five selected compounds (**1h**, and **1k–n**). The cells were maintained in Dulbecco’s modified Eagle’s medium (DMEM; Gibco, Life Technologies Inc., Frederick, MD, USA) supplemented with 10% (*v/v*) fetal bovine serum (FBS) and Antibiotic Antimycotic Solution (100 U penicillin, 100 μg/mL streptomycin and 0.25 μg/mL amphotericin B; Sigma-Aldrich, St. Louis, MO, USA) at 37 °C in a humidified incubator with a 5% CO_2_/95% air atmosphere.

In the cytotoxicity tests, SH-SY5Y cells were seeded at a density of 5 × 10^3^ cells/well in a 96-well plate and grown for another 24 h. Cells were then treated in triplicate with serial dilutions (100 μM, 50 μM, 25 μM, 12.5 μM, 6.25 μM, 3.125 μM) of selected compounds. After 48 h, the culture medium in each well was replaced with 100 μL of Resazurin working solution (44 μM). The fluorescent signal of the resorufin was monitored with Excitation λ = 530 nm and Emission λ = 590 nm by using an Envision 2104 Multi-label Microplate Reader (Perkin Elmer, Boston, MA, USA). Measurements were performed at time 0 and after 1 h of incubation. The percentage of viability was calculated after subtraction of the background (obtained by killing the cells), on the basis of the ratio between the fluorescence values of the cells incubated with a compound and the fluorescence values of cells incubated with the solvent (1% DMSO). The viability of cells incubated with 1% DMSO was considered to be equivalent to 100%. The statistical analysis was carried out with GraphPad Prism 6 (GraphPad Software, Inc., La Jolla, CA, USA) software using an unpaired t-test, and a *p*-value < 0.05 was considered statistically significant. Cytotoxicity (CC_50_) was determined from dose-response curves analyzed by using GraphPad Prism software.

### 3.3. Computational Methods

#### 3.3.1. Docking

We followed the validated procedure of [24]. In brief: substrates conformations were minimized with the AM1 method as implemented in MOPAC [25] and docked with AutoDock Vina [26] to the free protein (PDB id: 2IDZ) [27]. The docking cubic box was 30 Å side ad centered in the INH-adduct pocket (at coordinates 5.400, −32.000, 12.900 in the 2IDZ pdb frame). The docking was performed with an exhaustiveness of 500 and energy range of 50. Larger exhaustive energy ranges led to the same result. The calculations ran on 32 cpu and were performed on the MoNaLiSA cluster hosted at the University of Udine, Italy.

#### 3.3.2. Pocket Analysis

The target three-dimensional structure (PDB id: 2IDZ) was analyzed with the CavityPlus webserver [28] by employing both the Cavity utility for the pocket definition [29] and CavPharmer for the receptor-based pharmacophore modeling [30]. Data were visualized with VMD [31].

#### 3.3.3. Molecular Dynamics Simulations

We minimized the complex by first minimizing the protein side chains alone, then the whole protein and finally the whole system by constraining selected portions of the system. We placed the complex in a cubic box with a water layer of 0.7 nm and performed a second minimization. We used GROMOS 54a7 force field [32] and Simple Point Charge water. Ligand topologies were built with ATB [33]. We performed NVP and NPT equilibrations for 100 ps, followed by 250 ns NPT production run at 300 K. The iteration time step was set to 2 fs with the Verlet integrator and LINCS [34] constraint. We used periodic boundary conditions. All the simulations and their analysis were run as implemented in the Gromacs package v.2020.3 [35]. RMSDs and RMSF have been calculated from configurations sampled every 10 ps and as running averages over 100 sampled points. VINA scorings were calculated over configurations sampled every 100 ps and as running averages over 10 points. 2D ligand-protein interaction diagrams were generated with LigPlot+ [36]. The binding free energy was estimated with the MM-PBSA method, with the apolar solvation energy calculated as solvent accessible surface area (SASA) and default parameters, as implemented in the g_mmpbsa tool [37]. Simulations were run on Marconi (CINECA, Casalecchio di Reno, Italy).

## 4. Conclusions

In this work, we describe the synthesis of 14 novel hydrazide derivatives containing 1,3,4-oxadiazole core and the evaluation of their antimycobacterial activity.

With values of 8 µg/mL for five derivatives against the attenuated strain of *M. tuberculosis* H37Ra and 4 µg/mL against both the virulent strain of *M. tuberculosis* H37Rv and the pyrazinamide-resistant strain, the phenyl substitution in position 5 of the oxadiazole ring produced the best results within the series. Furthermore, the finding that none of the synthetic compounds demonstrated antifungal efficacy against a strain of *Candida albicans* supports their selectivity for mycobacteria. Interestingly, the majority of these compounds have a heterocycle ring linked to the carbohydrazide moiety.

By docking a selection of two compounds (**1a** and **1h**) to the biological target InhA, we performed a molecular modeling study to assess the potential mechanism of action of our molecules. In fact, owing to π-π stacking with the Phe41 residue, a simple unsubstituted phenyl ring appears both sterically and chemically appropriate as being more suited for the pocket shape and its pharmacophoric features. Moreover, the pharmacophore model (Figure 2a–c) revealed novel possibilities for additional functionalization. In particular, the decoration of the benzene ring with a hydrogen bond acceptor could lead to a ligand better suited for the identified pocket. Another option would be to cyclize the molecule by linking the hydrogen bond acceptor group to an aliphatic linker, connecting to the other benzene, and creating a macrocycle capable of blocking the large pocket entrance of InHA. An in silico analysis for the ADME properties of the most active derivatives was also carried out, and all the compounds were found to be potentially active if taken orally by humans. In this regard, the compounds could be chemically converted into the corresponding HCl salts to increase their water solubility.

## Data Availability

The data presented in this study are included in the manuscript and in Supporting Information. The corresponding author will provide further information upon request, to any qualified researcher.

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
