# Peer review of "Synthesis, Biological Evaluation and Computational Studies of New Hydrazide Derivatives Containing 1,3,4-Oxadiazole as Antitubercular Agents"

_ijms, 2022, doi:10.3390/ijms232315295_

Round 1

Reviewer 1 Report

The work "Synthesis, biological evaluation and computational studies of new hydrazide derivatives containing 1,3,4-oxadiazole as antitubercular agents" is a research article describing the design, synthesis and biological evaluation of novel aromatic and heteroaromatic hydrazide-based derivatives, proposed as antitubercular agents. Compounds were synthesized using published procedure and then evaluated for their antitubercular activity. The paper is well written, fluent in the description and the paragraphs consistently highlight the content of the article. I think that could be a step more in the discovery of new inhibitors. Unfortunately, novelty of the chemistry section is poor, and roughly described, it must be implemented carefully, considering the journal (IJMS is Q1 in Organic Chemistry!), anyway the biological data support a publication. Some critical points emerged during revision and, in my opinion, major correction are mandatory before accepting on International Journal of Molecular Sciences. I strongly encourage the author to improve the manuscript as follow.

Introduction

31-36 à the Introduction missed references about tuberculosis. Please add some.

39 à causative and causing resulted a repetition.

55-58 à “hydrazone moiety is also present in a large variety of compounds, including antiTB derivatives, not only because of hydrazone linkage but also due to various heterocyclic moieties brought together either from hydrazine, hydrazide or from a carbonyl compound”. This sentence doesn’t make much sense in my opinion. Please rearrange it. I think the authors wanted to stress out that activity of the compounds are related both to the hydrazone functionality and the chemical diversity of the carbonyl fragment. Is it correct?

61 à modify “at several” with “to several”, and “heterocyclics” with “heterocycles”.

Result and Discussion – Chemistry

The entire paragraph should be modified, because chemical synthesis is not described really well. First, paths a leads to a not isolated intermediate, right? So this should be removed and possibly paths a and b condensed in a single step operation, affording 2a,b. Scheme 1 could be aesthetically improved: the arrows should be aligned, avoiding that they overlap with the chemical structures, also to be aligned, perhaps formatted in ACS style. “Ar = various aromatic etc” is not so correct, because the authors only employed Phe, Pyr and Furyl aldehydes.

Modify with all the substituents. Ex. 1a, X = N, Ar = Ph.

“Reagentes” should be modified as “reagents”, and the entire sentence should be written after the text “Scheme 1. Synthetic route of title compounds 1a-n. Reagents and conditions: (a)…” Please check carefully Et3N, add in path d “appropriate aromatic or heteroaromatic aldehyde”.

Remove Ar-CHO from the arrow, and add in the Scheme 1 path d “appropriate aromatic or heteroaromatic aldehyde”.

69 à modify “oxazole” with “oxadiazole”.

79 à are the compounds new or already reported in the literature?

79 à modify “in solid form” with “as solids”

79 à TLC is not an adequate characterization methods. Remove it.

83-86 à do you think that stereochemistry could be important for the activity? Did you tried to separate the isomers?

Result and Discussion - Antimycobacterial activity

Obtained biological data showed weak antimycobacterial activity, compared to INH. To improve the consistency of the work it would be necessary to explain how these new compounds could lead to an improvement in existing therapies.

91 à “in vitro” and “M. tuberculosis” should be italic.

110 à Cl and Me are the only substituents on the aromatic ring investigated by the authors, therefore defining that the activity is not altered by electron attracting and donor groups is an unsupported finding. To affirm this, it would be necessary to consider a different panel of substituents with electronic effects on the ring and vary their position, therefore I suggest removing this sentence and limiting to a description of the variation in activity based on the available substituents.

Result and Discussion - In silico pharmacokinetic parameters

Are the compounds soluble in water? Based on the solvents used for NMR (DMF, DMSO, Py) I could deduce that the compounds manifest poor solubility in common organic solvents. The in silico pharmacokinetic analysis appears to be in contrast. For a compound to be administered orally, solubility in water is at least necessary. Have you thought about salifying N of pyridine?

Materials and methods - Chemistry

d-solvents should be reported as follow: CDCl3, DMSO-d6, DMF-d7.

213 à F254

 228 à check IUPAC name

The procedure should be implemented adding g, mol, equiv and mL where necessary. Yields should be written without decimals. Why 13C is not reported? Please add.

234-235 à Is it an extraction? Please explain.

241 à also for compound 2b should be implemented the procedure, because it’s a different molecule.

247 à please specify mL of ethanol.

248 à “Upon cooling a chromatographically 248 pure solid was collected by filtration” this sentence doesn’t make any sense. Please modify.

Compound 1c à carbon signals missing. Please check.

290 à modify “pirr” with “pyrr” or “pyrrole”

Compound 1f à a carbon signal is missing. Please check.

Compound 1i à check carbon signal, please.

357 à “e arom” modify please.

Reviewer 2 Report

The study done by Zampieri et al involved a design, synthesis and biological evaluation a library of 14 new hydrazide derivatives containing 1,3,4-oxadiazole unit followed by testing for antimycobacterial activity towards several mycobacterial strain including some resistant ones. The results obtained revealed that 5 compounds showed high antimycobacterial activity with MIC value of 8 µg/mL against M. tuberculosis H37Ra dormant strain and 2 derivatives proved to be effective against pyrazinamide-resistant strain with MIC of 4 µg/mL. The synthesized compounds were also tested against fungal C. albicans strain and showed no antimycotic activity proving a good selectivity profile, as well as a low cytotoxicity towards mammalian SH-SY5Y cells. Authors also performed a computational study in order to propose a plausible mechanism of action of the molecules towards the active site of InhA enzyme, and the results confirmed their hypothesis. The same active compounds were in silico predicted for ADME properties and all proved to be potentially orally active in humans, The study is very interesting and have great scientific value; the manuscript is well written; the results are clearly presented; however, the introduction part needs improvement. The manuscript needs some minor corrections.

1.     Materials and methods. Line 388, replace 108 to 108.

2.     Materials and methods. Line 409, replace 30 mL to 30 µL.

3.     Materials and methods. Line 414, the authors stated that Isoniazid was used as reference compound, However, in my opinion more than one reference compound should be used since INH-R ATCC35822 strain was included in the test which is resistant to Isoniazid.

4.     Materials and methods. Line 425, replace 100 αl inoculum of C. albicans to 100 µl.

5.     Materials and methods. Line 432, replace the sentence “This cell-permeable dye is reduced to resorufin by aerobic respiration of metabolically active cells to resorufin” by “This cell-permeable dye is reduced by aerobic respiration of metabolically active cells to resorufin”.

6.     Materials and Methods. Line 445, add the proper concentration ranges. Cells were then treated in triplicate with serial dilutions (100 µM, 50µM, 25 µM, 12.5 µM, 6.25 µM, 3.125 µM) of selected compounds.

7.     Results. Line 91 M. tuberculosis should be italic, please revise through all the manuscript.

8.     Results. Table S1. Add concentration unit, Figure S1. Assign a different colour for compound 1h, change either graph line thicknesses or y axis scale or add the percentage viability data to each point on the graph to differentiate between compounds 1h and 1n.

Round 2

Reviewer 1 Report

I really thank the authors for considering my suggestions!

Good luck for your publication

Author Response

Dear reviewer thank you very much. Your helpful comments and suggestions, which we gladly adopted, indeed contributed to improving the final quality of our manuscript.